# Current Progress in the Development of mRNA Vaccines Against Bacterial Infections

**DOI:** 10.3390/ijms252313139

**Published:** 2024-12-06

**Authors:** Alina Khlebnikova, Anna Kirshina, Natalia Zakharova, Roman Ivanov, Vasiliy Reshetnikov

**Affiliations:** Translational Medicine Research Center, Sirius University of Science and Technology, Sochi 354340, Russia; alina.s.x@yandex.ru (A.K.);

**Keywords:** RNA vaccine, bacterial infection, *M. tuberculosis*, *P. aeruginosa*, *L. monocytogenes*, *S. pyogenes*, *B. burgdorferi*

## Abstract

Bacterial infections have accompanied humanity for centuries. The discovery of the first antibiotics and the subsequent golden era of their discovery temporarily shifted the balance in this confrontation to the side of humans. Nevertheless, the excessive and improper use of antibacterial drugs and the evolution of bacteria has gotten the better of humans again. Therefore, today, the search for new antibacterial drugs or the development of alternative approaches to the prevention and treatment of bacterial infections is relevant and topical again. Vaccination is one of the most effective strategies for the prevention of bacterial infections. The success of new-generation vaccines, such as mRNA vaccines, in the fight against viral infections has prompted many researchers to design mRNA vaccines against bacterial infections. Nevertheless, the biology of bacteria and their interactions with the host’s immunity are much more complex compared to viruses. In this review, we discuss structural features and key mechanisms of evasion of an immune response for nine species of bacterial pathogens against which mRNA vaccines have been developed and tested in animals. We focus on the results of experiments involving the application of mRNA vaccines against various bacterial pathogens in animal models and discuss possible options for improving the vaccines’ effectiveness. This is one of the first comprehensive reviews of the use of mRNA vaccines against bacterial infections in vivo to improve our knowledge.

## 1. Introduction

Bacterial infections are becoming an increasingly serious public health problem every year, and the number of deaths caused by the complications of bacterial infections is growing steadily. The problem of increased mortality from bacterial infections and negative prognosis are mostly due to two main reasons.

Firstly, there is the increasing resistance of major species of pathogenic bacteria to existing antibacterial drugs, the main discoveries of which occurred in the 1940s–1960s [1]. The excessive and improper use of antimicrobials allow microorganisms to quickly evolve, acquiring resistance and thereby increasing the risk of development and spread of diseases among people [2,3]. In addition, antibiotics are used not only to treat or prevent diseases but also in agriculture, and this situation also contributes to the spread of resistant strains of bacteria. The second important reason is the few developments in the field of new-generation antibacterial drugs [4]. Now, we can distinguish three main areas in this field of medical chemistry: the design of antibiotics with a fundamentally new mechanism of action, the improvement of the effectiveness of old ones through various modifications, and the development of so-called potentiators: substances that do not have antibacterial activity on their own but when administered together with antibiotics enhance their effectiveness [4,5].

Vaccination is an efficient and alternative approach to prevent the spread of bacterial infections and reduce the associated mortality. Vaccines against tuberculosis, pneumococcal and meningococcal infections, diphtheria, whooping cough, tetanus, and other infections are already used successfully. On the other hand, these vaccines are not effective in all cases, and therefore the design of new alternative highly effective vaccines can improve the prevention of bacterial complications.

mRNA vaccines as third-generation vaccines—in contrast to live attenuated, inactivated vaccines, and second-generation (subunit) vaccines—show a higher speed of research and development, offer a possibility to flexibly select antigen composition, and entail a relatively inexpensive and scalable production process. mRNA vaccines enable the “fine-tuning” of the optimal balance between humoral and cellular immunity, particularly Th1/Th2 immunity. The arsenal of mRNA vaccines includes a wide range of tools for modulating an immune response. Aside from the possible use of classic adjuvants, genetic adjuvants can be applied here, which have shown their effectiveness [6]. Another opportunity with mRNA vaccines is the creation of multiepitope vaccines that encode only individual epitopes of target antigens, thereby possibly increasing the effectiveness of such modalities [7] and reducing potential adverse effects by eliminating unwanted protein sequences. In silico tools allow for the selection of optimal CTL, HTL, and LBL epitopes and for the modeling of a potential immune response to vaccination [8,9].

The development of mRNA-based vaccines for the prevention of bacterial infections is a promising research avenue, but in contrast to viruses, the task of choosing target antigens or their combination is more complicated. Understanding which bacterial antigens are most important for providing an immune response will accelerate the development of a rational design of bacterial mRNA vaccines. Surface proteins, lipo-polysaccharides and toxins can be considered as important antigens involved in the pathogenesis. Bacteria can act as intracellular or extracellular pathogens and be of different types (Gram-positive or Gram-negative). Depending on the type of bacteria, different strategies are required in vaccine development. In addition, many bacteria, such as *Mycobacterium tuberculosis*, have developed effective mechanisms for avoiding an immune response [10].

The development and implementation of effective vaccines in combination with antibiotic therapy play an important role in the control of pathogens, especially antibiotic-resistant microorganisms, as reflected in recent reviews on this topic [11,12]. However, the number of reviews on the use of mRNA vaccines against bacterial infections is limited [13]. The aim of this review was to summarize the existing data on successes in the creation of mRNA vaccines against various bacterial infections and to identify possible avenues of research for improving their effectiveness. A special feature of our review is a detailed analysis of the molecular mechanisms of the pathogenesis of bacterial infections, which allows us to comprehensively assess the prospects and challenges associated with the use of mRNA vaccines for their prevention.

## 2. Statistics on Cases of Illnesses and Deaths Due to Infections Caused by Bacteria

In 2019, of the 13.7 million deaths associated with infections, 7.7 million were caused by 33 bacterial pathogens that are either resistant or sensitive to antibiotics. These pathogens were responsible for 13.6% of all deaths. According to estimates from the World Health Organization (WHO), in 2019, 4.95 million deaths could be attributed to antibiotic resistance [14].

In 2017, the WHO publicized a priority list of key bacteria having antibiotic resistance [15]. This list has become the basis for activities related to the surveillance and control of antibiotic resistance. Much later, after the COVID-19 pandemic, this list has undergone substantial changes. The new 2024 list [16] includes 15 families of pathogens resistant to antibiotics. Nonetheless, there are other pathogens, such as those inducing plague and listeriosis, that are not on this list. But cases of these infections continue to occur from time to time. but outbreaks of these infections occur from time to time.

The WHO has also compiled the ESKAPE list pathogens (*Enterococcus faecium*, *Staphylococcus aureus*, *Klebsiella pneumoniae*, *Acinetobacter baumannii*, *Pseudomonas aeruginosa,* and *Enterobacter* species); these are the most virulent pathogens featuring multidrug resistance, against which the development of antimicrobials is a priority according to the WHO and the Centers for Disease Control (CDC) [17].

In this present review, we focus on the results of in vivo studies about the use of mRNA vaccines against bacterial diseases. The list of nine pathogens for which such vaccines are being developed and the current data on existing vaccines against these pathogens are presented in Table 1.

### 2.1. M. tuberculosis

Tuberculosis caused by *M. tuberculosis* is one of the leaders in mortality among infectious diseases. According to the WHO, in 2022, the prevalence of tuberculosis was 10.6 million people, and the mortality rate was 1.3/10.6 [19]. Although there is an approved tuberculosis vaccine (BCG), one of its drawbacks is decreased effectiveness in adults against the most common type of tuberculosis, pulmonary [20]. Since the year 2020, the number of patients with tuberculosis showing either resistance to first-line antibiotic therapy or multidrug resistance continues to grow [19]. Therefore, today, there is an urgent need for an effective preventive vaccine against tuberculosis.

#### 2.1.1. Features of the Structure and Immune Evasion

Infection with the tuberculosis pathogen occurs when aerosol particles carrying *M. tuberculosis* enter the respiratory tract through the nose or mouth. After reaching the alveoli, *M. tuberculosis* is phagocytosed by alveolar macrophages and dendritic cells [21,22]. *M. tuberculosis* has protein and nonprotein virulence factors in its arsenal, which are described in detail in a recent review [23]. Due to various mechanisms of virulence and immune-system evasion, *M. tuberculosis* is able to survive in various intracellular environments of myeloid cells. In phagocyte lysosomes, *M. tuberculosis* can avoid degradation by inhibiting maturation and acidification of the phagolysosome and can suppress antigen processing for presentation to cytotoxic T lymphocytes (CTLs). Bacterial proteins, by reducing pH, prevent the accumulation of vacuolar enzymes, ATP, and GTP, thereby affecting the maturation of phagocytes. These proteins include CFP10 (culture filtrate protein), ESAT6 (early secretory antigen 6), and ATF1/2 [24]. *M. tuberculosis* is capable of producing substances that reduce the expression of PI3P, which is necessary for endosome–lysosome fusion and for the transport of proteins from the endosome for antigen presentation [25].

If the pathogen is not eliminated at the initial stage, then infected macrophages migrate to the interstitium, where the infection cycle continues owing to the phagocytic activity of resident macrophages. Macrophages also migrate to draining lymph nodes along with infected dendritic cells, where they activate T and B cells, which then return to the site of infection [10]. The T and B cells as well as attracted inflammatory monocytes, dendritic cells, neutrophils, and natural killers form a structure that is characteristic of tuberculosis and is called granuloma [26]. Granuloma can restrain the progression of the infection in combination with an effective adaptive response. At the same time, granuloma is able to maintain the viability of *M. tuberculosis* in a dormant state, thus limiting the adaptive immune response [23]. 

Humoral immunity against *M. tuberculosis* is quite effective even with a short-term presence of the pathogen outside the cell. Opsonization can enhance the responses of the innate immune system against *M. tuberculosis* [27]. The phagocytic function of macrophages and the cytotoxicity of natural killers can be enhanced by antibodies against *M. tuberculosis* [28]. It has been shown that as a result of vaccination, IgM antibodies specific to *M. tuberculosis* can provide effective protection against *M. tuberculosis* [29].

#### 2.1.2. Non-mRNA Vaccines Against *M. tuberculosis*

To combat tuberculosis, the creation of new antituberculosis vaccines is required. Many candidate vaccines are currently in clinical and preclinical trials [30,31]. Some of them involve strategies aimed both at modifying existing BCG [32,33,34,35] and at developing other types of vaccines, including inactivated, attenuated, protein subunit, and nucleic-acid-based vaccines [30]. 

#### 2.1.3. mRNA Vaccines Against *M. tuberculosis*

The mRNA vaccine platform was first used against tuberculosis in 2004 by Tian Xu et al. [36] (Table 2). Of note, self-amplifying RNA (saRNA) expressing the MPT83 antigen of *M. tuberculosis* was used as the RNA platform there. The RNA was administered naked, without the use of delivery systems. Immunization with the mRNA vaccine based on MPT83 resulted in the formation of humoral (IgG titer) and cellular immunity (secretion of IFN-γ by splenocytes in response to a specific stimulus) in mice. After infection of the vaccinated mice with *M. tuberculosis* H37Rv, a decrease in the bacterial load in the lungs was observed 4 weeks after the last immunization.

Only 6 years later, in 2010, another study was conducted by the group of Dr. Lorenzi [37] regarding mRNA encoding the Hsp65 protein of *M. leprae* for the prevention of tuberculosis. In that work, mRNA was also administered naked but intranasally. It was found that after the intranasal administration of the mRNA vaccine, RNA entered the dendritic cells of mice lungs. Immunization with the mRNA vaccine based on the sequence of Hsp65 led to a decrease in the bacterial load in the lungs and prevented their damage. Vaccination with mRNA–Hsp65 gave rise to a Th1-directed immune response.

After the year 2010, only in 2023, new experimental research articles appeared dealing with the use of mRNA vaccines against tuberculosis. In most of these studies, an RNA delivery system was used. The research group of Larsen et al. [38] used—as an antigen in a saRNA vaccine—fusion protein ID91 consisting of four fused proteins of *M. tuberculosis*: Rv3619 (esxV; ESAT6-like protein), Rv2389 (RpfD), Rv3478 (PPE60), and Rv1886 (Ag85B).

Vaccination with ID91 saRNA via a delivery system in the form of a nanostructured lipid carrier elicited a cellular and humoral immune response [38]. The administration of ID91 saRNA also provided prophylactic protection because the immunized mice showed reduced bacterial load in the lungs at 3 weeks after infection with a low dose of *M. tuberculosis* H37Rv. The researchers also tested different prime–boost strategies and found that the use of the mRNA vaccine as a prime and a protein vaccine as a boost induced a significant reduction in bacterial load in the lungs.

The same group of researchers assessed the protective properties of the same vaccines within heterologous vaccination against *Mycobacterium avium* [39]. *M. avium* is a slow-growing mycobacterium, and there is no approved vaccine against this pathogen yet. Those authors investigated homologous (RNA–RNA and protein–protein) and heterologous (RNA–protein and protein–RNA) prime–boost strategies against *M. avium* [39]. The heterologous prime–boost RNA–protein vaccination strategy showed superior results and caused the formation of cellular and humoral responses against *M. avium* antigens, with significantly elevated levels of interleukin 1β (IL-1β), IL-6, and IL-10 and titers of IgG, IgG1, and IgG2c antibodies specific to ID91 and Ag85B in mouse serum samples. Additionally, the heterologous prime–boost RNA–protein vaccination strategy elicited sustainable CD4^+^ Th1 immune responses to all four components of ID91 as well as increased numbers of CD4^+^CD44^+^ splenocytes producing CD154, IFN-γ, IL-2, and TNF. The heterologous prime–boost RNA–protein vaccination strategy also resulted in the formation of a CD8^+^ T-cell response, which manifested itself in the induction of the expression of proinflammatory cytokines IFN-γ, IL-2, and TNF by CTLs [39]. After the infection of mice (with *M. avium*) vaccinated with the heterologous prime–boost RNA–protein or protein–RNA strategy, a decrease in bacterial loads within the lungs and spleen was observed [39]. Thus, the strategies of combining an RNA vaccine with a protein vaccine may prove to be promising tools for the prevention of infectious diseases.

In another work on an mRNA vaccine against *M. avium* [40], investigators assessed the efficacy of an mRNA vaccine encoding amino acid sequences of antigens Ag85B and Hsp70 and formulated with QTAP (detergent Quil-A and DOTAP) nanoparticles. Immunization with QTAP–mRNA enlarged the population of CD4^+^ T cells in the lungs that produce Th1 (IFN-γ, TNF, and IL-2) and Th17 (IL-17A) cytokines but not in the number of CD8^+^ T cells. Immunization with QTAP–mRNA significantly reduced the bacterial load in the lungs and spleen after infection initiation.

In our series of studies, we tested an mRNA vaccine containing a sequence of secretory protein ESAT6. In the first study [7], a comparison of T-cell responses was performed in mice immunized with an mRNA vaccine encoding the full-length antigen or a multiepitope antigen of *M. tuberculosis*. We noticed that the number of IFN-γ-secreting cells among the splenocytes of immunized mice in response to a specific stimulus was higher in the mice vaccinated with the multiepitope version of the vaccine compared with the full-length ESAT6 version.

In another research article, we optimized sequences of the 5′ untranslated region of the multiepitope mRNA vaccine [41]. The best version (5′-TPL-mEpitope-mRNA1273-3′) gave rise to cellular and humoral immunity and diminished the bacterial load in the lungs. Furthermore, the immunization with 5′-TPL-mEpitope-mRNA1273-3′ improved the survival rate of the mice [42]. Nevertheless, compared to BCG, the efficiency of this mRNA vaccine was lower in both adaptive and protective immunity. Subsequently, we also optimized the structure of the synthetic analog of the cap, thus increasing the efficiency of the adaptive response caused by the mRNA vaccine; this response became stronger than that in the group of mice vaccinated with BCG; however, it was not possible to achieve an enhancement of the protective properties of the vaccine against infection with *M. tuberculosis* [43]. Consequently, these papers underscore that the use of multiepitope types of mRNA vaccines and the optimization of their regulatory regions can improve their efficacy.

Recently, Dr. Lukeman and colleagues [44] have developed a mRNA vaccine, mRNA^CV2^, encoding a fusion protein (CysVac2) of two mycobacterial antigens: secreted immunodominant Ag85B and CysD (a component of the sulfur assimilation pathway; this protein is overexpressed at chronic stages of the infection). Those authors demonstrated that immunization with the mRNA vaccine mRNA^CV2^ leads to the formation of cellular and humoral immune responses. The number of CD4^+^ T cells secreting cytokines characteristic of the Th1 pathway was significantly higher in animals vaccinated with mRNA^CV2^ compared to the controls. Those authors showed that the vaccination also generates protective immunity; the bacterial load in the lungs was significantly lower after infection with the *M. tuberculosis* strain H37Rv. Thus, the mRNA^CV2^ vaccine ensures pronounced cellular and humoral responses and has protective properties. Moreover, the authors of Ref. [44] demonstrated the ability of mRNA^CV2^ to enhance existing immunity after BCG vaccination. Six months after vaccination with BCG, mice were revaccinated twice with mRNA^CV2^. Before the boost immunization with mRNA^CV2^, the mice vaccinated with BCG showed only minor residual responses of CD4^+^ T cells compared to unvaccinated mice. By contrast, after the boost vaccination with mRNA^CV2^, a significant increase in the frequency of IFN-γ^+^ and TNF^+^CD4^+^ T cells was observed compared to groups receiving only BCG or mRNA^CV2^.

After a challenge with *M. tuberculosis*, mice immunized with both BCG and mRNA^CV2^ were the only group that demonstrated a significant reduction in bacterial loads in both the lungs and spleen compared to the unvaccinated mice [44]. Thus, mRNA^CV2^ is able to enhance the immune protection obtained in response to early vaccination with BCG and can be considered a booster vaccine.

**Table 2 ijms-25-13139-t002:** Using RNA vaccines to prevent bacterial and parasitic infections in vivo (in animals and humans).

Pathogen	Type of Vaccine	Delivery System, Route	Antigen(s)	Uridine Analogs	Additional Cassette Elements	Dose/Treatment Regimen	Animals	Results: Adaptive Immunity	Results: Protective Immunity	Reference
** *M. tuberculosis* **	saRNA	Naked, intramuscularly	MTP83	-	RNA replicase from the Sindbis virus	50 μg, four times every 3 weeks	♀ BALB/c mice	IgG ↑, IFN-γ of splenic cells ↑	Bacterial load in lungs ↓	[36]
	Linear RNA	Naked, intranasally	Hsp65 (*M. leprae*)	-	-	10 μg	♀ BALB/c mice	Specific Th1 cytokine production (IFN-γ ↑, TNF ↑ of splenic cells)	Bacterial load in lungs ↓	[37]
	saRNA	NLC, intramuscularly	Fusion protein ID91: Rv3619 (esxV; ESAT6-like protein), Rv2389 (RpfD), Rv3478 (PPE60), and Rv1886 (Ag85B)	-	nsPs from VEEV	1.0 μg, once or twice 3 weeks apart	♀ C57BL/6 mice	Ex vivo IFN-γ ↑ in PPD^+^ human PBMCs, CD8^+^ T-cell proliferation and cytokine induction, IL-2 and TNF-producing CD8^+^ CD44^+^ T cells, IgG ↑	Bacterial load in lungs ↓	[38]
	Linear RNA	LNPs, intramuscularly	Five epitopes of ESAT6 (rv3875)	-	-	50 μg, twice 3 weeks apart	♀ C57BL/6JCit and I/StSnEgYCit mice	IFN-γ of splenic cells ↑, development of delayed-type hypersensitivity reaction	Bacterial load in lungs ↓	[41]
	Linear RNA	LNPs, intramuscularly	Five epitopes of ESAT6 (Rv3875) or full-length Rv3875	-	-	50 µg, twice 4 weeks apart	♂ C57BL6/J mice	IFN-γ of splenic cells ↑	-	[7]
	Linear RNA	LNPs, intramuscularly	Five epitopes of ESAT6 (Rv3875)	-	-	50 μg, twice 3 weeks apart	♀ C57BL/6Cit and I/StSnEgYCit mice	IFN-γ of splenic cells ↑, development of delayed-type hypersensitivity reaction, IgG ↑	Bacterial load in lungs ↓, survival % ↑	[42]
	Linear RNA	LNPs, intramuscularly	Five epitopes of ESAT6 (Rv3875)	-	-	50 μg, twice 3 weeks apart	♀ C57BL/6Cit and I/StSnEgYCit mice	IFN-γ of splenic cells ↑, development of delayed-type hypersensitivity reaction, IgG ↑	No protective effect	[43]
	Linear RNA	LNPs, intramuscularly	Five epitopes of ESAT6 (Rv3875	-	-	5 μg or 15 μg twice 2 weeks apart	♀ Wistar rats	IFN-γ of splenic cells ↑	-	[45]
	Linear RNA	LNPs, intramuscularly	Fusion protein CysVac2 (Ag85B and CysD)	N1-mΨ	SEAP signal peptide	5 μg, three times 3 weeks apart	Female C57BL/6 mice	Th1 immune response, Ag85B:I-Ab^+^ CD4^+^ ↑, IgG1 and IgG2c ↑	Bacterial load in lungs ↓	[44]
** *M. avium* **	saRNA	LION™, intramuscularly	Fusion protein ID91: Rv3619 (esxV; ESAT6-like protein), Rv2389 (RpfD), Rv3478 (PPE60), and Rv1886 (Ag85B)	-	nsPs from VEEV	1.0 μg, two times 4 weeks apart	♂ and ♀ C57BL/6 bg/bg mice	Anti-ID91 IgG1 ↑ and IgG2c ↑, anti-Ag85B IgG1 ↑	Bacterial load in lungs and liver ↓, lung damage ↓	[39]
** *M. avium* ** ** subspecies** ** * hominissuis* **	Linear RNA	Nanoadjuvant system QTAP, subcutaneously	Ag85B, Hsp70	Ψ	-	15 μg, three times 5 weeks apart	C57BL/6 mice	In vitro: CD80- and CD86-expressing macrophages ↑, NLRP3 ↑, NF-κB ↑, MyD88 ↑, cell survival ↑; ↑ lung CD4^+^ T cells producing Th1 (IFN-γ, TNF, and IL-2) and Th17 (IL-17A) cytokines	Bacterial load in lungs and spleen ↓	[40]
** *B. burgdorferi** **	Linear RNA	LNPs, intramuscularly	OspA	N1-mΨ	-	3 μg, single dose or twice 4 weeks apart	♀ BALB/c mice	↑ CD4^+^ and CD8^+^ T cells expressing Th1-associated cytokines (IFN-γ, IL-2, and TNF) in the spleen, CXCR5^+^PD-1^+^ Tfh cells ↑ and FAS^+^GL7^+^ GC B cells ↑ in LN, CD38^+^GL7^–^ MBCs ↑ in the spleen and B220^+^CD138^+^ LLPC ↑ in bone marrow, ↑ IgG1, IgG2a, and IgG2b in bone marrow	Bacterial load in bladder, heart, skin ↓	[46]
	Linear RNA	LNPs, intradermally	Salp14, TSLPI, Salp10, Salp15, Salp16A, Salp17, Salp25A, Salp25B, Salp25C, Salp25D, Salp26A, TIX5, P32, P11, Salp12, SG09, SG10, SG27, and IsPDIA3	N1-mΨ	IL-2 signal peptide	50 μg, three times every 4 weeks	♀ guinea pigs	Antibody response to 10 of 19 antigens, ↑ tick immunity, IFN-γ ↑, TNF ↑, CXCL10 ↑, IL-2 ↑, IL-4 ↑, IL-8 ↑	60–100% of pigs not infected	[47]
	Linear RNA	LNPs, intradermally	Salp14, TIX5, P32, Salp26A, TSLPI, Salp15, Salp25D, P11, SG27, SG09, SG10, and IsPDIA3	N1-mΨ	-	3 μg, three times every 4 weeks	♀ guinea pigs	↑ tick immunity, serum *IgG* ↑, intense and earliest erythema at the tick bite site	-	[48]
	Linear RNA	LNPs, intradermally	Salp14	N1-mΨ		20 μg, three times every 4 weeks	Female Hartley guinea pigs	*IgG* ↑	*-*	[49]
* **B. pertussis** *	Linear RNA	LNPs	PTX-S1, PRN, FHA3, FIM2/3, RTX, BRKA, TCFA, SPHB1, DT, and TT	N1-mΨ	IgG kappa or bovine prolactin, native signal sequence from *Corynebacterium diphtheriae* for DT its own signal sequence	10 µg	C57BL/6J mice	IgG ↑ Responses of CD4^+^ IFN-γ^+^ Th1, CD4^+^CXCR3^+^ TEM and TCM ↑ CD8^+^CXCR3^+^ TEM and TCM	Bacterial load in lungs and trachea ↓	[50]
						10 µg	♀ Sprague Dawley rats	IgG ↑		[51]
** *S. pyogenes* **	saRNA	CNE	SLOdm	-	MF59	15 µg (three times, on days 1, 21, and 35)	♀ CD-1 mice	IgG ↑ Th1 immune response	Survival % ↑	[52]
** *S. agalactiae* **	saRNA	CNE	BP-2a	-	MF59 signal peptide (murine Ig κ-chain leader sequence)	15 µg (three times, on days 1, 21, and 35)	♀ CD-1 mice	IgG ↑ Th1 immune response	Protection of offspring from lethal infection	[52]
	Linear RNA	LNPs	30 N-terminal M peptides, peptides Spa	-	C-terminal transmembrane anchor (CtTM)	100 µg, three times every 4 weeks	♀ New Zealand white rabbits	IgG2a, IgG2b ↑	-	[53]
** *P. aeruginosa* **	Linear RNA	LNPs, intramuscularly	PcrV OprF-I	N1-mΨ	Al(OH)_3_ tPAss, 6xHis tag	5 and 25 μg, twice every 3 weeks	BALB/c mice	IFN-γ-producing splenic cells ↑, IgG ↑ CD4^+^ Th1 ↑	Survival % ↑, bacterial load in burn and systemic-infection models ↓	[54]
	Linear RNA	LNPs, intramuscularly	PcrV	N1-mΨ	tPAss, trimerization domain	10 μg, three times every 2 weeks	♂ ICR mice	Lung IgA ↑, serum IgG ↑,	Survival % ↑, bacterial load in lungs ↓	[55]
** *L. monocytogenes* **	Linear RNA	Cationic LNPs	OppA, LLO	N1-mΨ	α-GC	10 µg	♀ C57BL/6J mice	IFN-γ ↑, specific CD8^+^ T-cell responses	-	[55]
** *C. trachomatis* **	saRNA	CAF	MOMP	-	-	5 μg	♀ BALB/c mice	IgG ↑ IFN-γ^+^ T-cell response ↑	-	[56]
** *Y. pestis* **	saRNA	LNPs	LcrV (V), F1	-	-	1 or 5 μg, twice 4 weeks apart	♀ BALB/c, OF1 mice	IFN-γ-producing splenic cells ↑, IgG ↑	Survival % ↑, bacterial load in spleen↓	[57]
	Liner RNA	LNPs	F1 (caf1)		Conjugation with hFc (human)	5 µg, three times every 2 weeks	♀ C57BL/6JOlaHsd mice	IgG ↑, anti-F1 IgG ↑	Full protection against lethal infection	[58]
***Rhodococcus equi*** **	Linear RNA	LNPs, intramuscularly	VapA	-	-	300 µg	Horses	IFN-γ ↑, IgG_4/7_, IgG_1_ ↑, IgA ↓	-	[59]

NLC: nanostructured lipid carrier; LNP: Lipid nanoparticle; LIONTM: Lipid InOrganic Nanoparticle; CNE: Cationic nanoemulsion; CAF: Cationic adjuvant formulation; PBMC: Peripheral blood mononuclear cell; LN: Lymph node; nsPs: non-structural proteins; * are not bacterial antigens; ** *Rhodococcus equi*: vaccination of animals; ♀: female; ♂: male; ↑: increase; ↓: decrease.

### 2.2. Borrelia burgdorferi

The *B. burgdorferi sensu lato* group of bacteria (hereafter *B. burgdorferi*) includes four species that are pathogenic to humans and can cause Lyme disease: *B. burgdorferi sensu stricto, B. garinii, B. afzelii,* and *B. bavariensis* [60]. *B. burgdorferi* is a Gram-negative, spiral-shaped spirochete bacterium that is transmitted via bites of *Ixodes* ticks [60,61]. Natural hosts of *B. burgdorferi* are mainly various small mammals, especially rodents, in which infection does not lead to a disease [60,61]. As a result of the infection of a person by *B. burgdorferi*, a local infection of the skin (erythema migrans) develops initially. If untreated, the infection spreads to other parts of the body, as manifested most often by neurological aberrations. As a consequence, the bacterial cells manage to survive for long periods in one or two localized niches, most often resulting in arthritis, carditis, neurological disorders, or chronic atrophic acrodermatitis [60]. *B. burgdorferi* does not excrete toxins or proteases, and therefore, most manifestations of Lyme borreliosis in humans are the result of inflammation caused by the body’s immune response to surface antigens, microbiota, and tick saliva after a bite [62].

#### 2.2.1. Features of the Structure and Immune Evasion

*B. burgdorferi* is covered by two lipid bilayers, an outer and an inner one, between which flagella are located in the periplasmic space, which are attached to the poles of the cell and extend over the entire surface of *B. burgdorferi*, thereby giving it a spiral shape [63]. The presence of seven to eleven flagella allows *B. burgdorferi* to easily move within tick saliva and in the viscous extracellular matrix of the mammalian dermis. The location of highly conserved and immunogenic flagella (hidden under the outer membrane) protects them from recognition by the host’s immune system [62].

The surface of *B. burgdorferi* is devoid of outer-membrane lipopolysaccharides; instead, it contains multiple lipoproteins, the expression of which is regulated depending on environmental conditions and allows *B. burgdorferi* to evade the immune response [64]. After attachment of the tick and the start of its feeding, changes in temperature, nutrient levels, and pH trigger the transformation of *B. burgdorferi* [65]. One of the manifestations of the transformation is the expression of RNA polymerase alternative σ-factor RpoS and *Borrelia* oxidative stress regulator (BosR), which are responsible for the regulation of multiple genes encoding surface lipoproteins. In particular, the two regulators participate in the “switch” of expression from OspA (ensures attachment of *B. burgdorferi* to the tick midgut) to OspC (RpoS activates transcription of *ospC*, whereas BosR inhibits transcription of *ospA*) [66]. Being located on the surface of the outer membrane of *B. burgdorferi*, OspC promotes bacterial colonization by evading immune responses because it can bind the Salp15 protein (from tick saliva), which has immunosuppressive properties [67]. The transcription of *ospC* is then replaced by *vlsE* transcription, apparently when the bacterium again needs to evade the immune response consisting of antibodies after the progression of the infection [66]. By means of VlsE from 31 strains of *B. burgdorferi* as an example, the capacity of *B. burgdorferi* for antigenic variability has been investigated. *vlsE* is located in the lp28-1 plasmid together with a series of 15 *vls* cassettes, whose sequences are highly similar to *vlsE*. It has been reported that parts of the expressed *vlsE* locus can be replaced by sequences of silent *vls* cassettes as a consequence of recombination, thereby giving variable VlsE proteins. The rearrangement of *vlsE* is well pronounced in immunocompetent mice but is considerably reduced in SCID mice and is not observed in ticks and in vitro. Infection of immunocompetent mice with *B. burgdorferi* that is devoid of plasmid lp28-1 is difficult but is unchanged (typical) in SCID mice. These data indicate that one of the strategies of *B. burgdorferi* for evading a host immune response is antigenic variability. In humans, the role of VlsE in immune evasion remains to be elucidated, but the active production of VlsE-specific antibodies is seen at the early stages of Lyme disease [68]. Although not all *B. burgdorferi* strains contain the lp28-1 plasmid, they do carry the *vls* locus in other linear plasmids [69].

Another strategy of *B. burgdorferi* for evading an immune response is its resistance to the complement system. Depending on its strain, *B. burgdorferi* has various amounts of lipoproteins, collectively called complement regulator-acquiring surface proteins (CRASPs), which are capable of binding to complement regulatory factor H with the factor H-like protein and factor H-related proteins, thereby inactivating C3b, and thus preventing the launch of the complement pathway and therefore the destruction of the bacterial cell [70,71,72].

Infection with *B. burgdorferi* is treatable with antibiotics, but sometimes the disease is difficult to detect at early stages [73]. Moreover, even after antibiotic therapy, post-treatment Lyme disease syndrome may occur, which includes joint and muscle pain, fatigue, headache, and neurocognitive problems [74]. In this regard, the development of a prophylactic vaccine is necessary.

#### 2.2.2. Non-mRNA Vaccines Against *B. burgdorferi*

Currently, there is no approved vaccine against the development of Lyme borreliosis. In 1998, the FDA approved LYMErix: an OspA protein vaccine with proven efficacy and safety. In the same year, however, a research article came out indicating that OspA contains an epitope homologous to a peptide in hLFA-1, and that this similarity can induce cross-reactive autoimmune processes [75]. Although OspA has been subsequently found to not be able to cause these reactions, sales of the vaccine have declined, and in 2002, it was withdrawn from the market [76]. On the other hand, the approved vaccine for dogs against Lyme disease is also based on OspA [77]. The only candidate vaccine against Lyme disease currently in a phase III clinical trial (NCT05477524) is multivalent recombinant vaccine VLA15, also based on OspA [78].

#### 2.2.3. mRNA Vaccines Against *B. burgdorferi*

At present, there are several preclinical studies evaluating the mRNA platform as a vaccine against infection by *B. burgdorferi*. Matthew Pine et al. [46] have employed the aforementioned OspA as an antigen in their mRNA vaccine. It was shown in their work that on the first day after immunization, the number of neutrophils at the injection site significantly goes up; on the third day, so do the number of dendritic cells, macrophages, and monocytes; and on the fifth day, the numbers of these cells diminish. The OspA mRNA vaccine delivered via lipid nanoparticles (LNPs) was found to increase the levels of antigen-specific CD4^+^ and CD8^+^ T cells expressing Th1-associated cytokines (IFN-γ, IL-2, and TNF) in the spleen. The number of antigen-specific CD38^+^GL7^–^ mononuclear blood cells of the spleen and B220^+^CD138^+^ LLPCs of bone marrow from the immunized mice were significantly higher than those in the controls. Furthermore, elevated levels of different IgG subtypes (IgG1, IgG2a, and IgG2b) were detectable in the bone marrow. Taken together, this evidence points to the formation of robust humoral immunity after one-time immunization. After two-time immunization, the specific IgG response to OspA mRNA–LNP was shown to persist for up to 24 weeks.

After the infection of immunized mice with *B. burgdorferi*, the bacterial load is reported to be significantly reduced in the bladder, heart, skin, and serum compared to controls, but not in the joints [46]. Immunization with mRNA–LNP in that study provided protection against *B. burgdorferi* in eight out of ten mice.

Narasimhan et al. have demonstrated that immunity to tick bites can suppress infection with *B. burgdorferi* in a guinea pig model [79,80]. In another paper [47], an mRNA vaccine (19ISP) was developed that contains the nucleotide sequences of 19 tick salivary-gland proteins. Those researchers tested whether immunization with such an mRNA vaccine would provide immunity to ticks and prevent transmission of *B. burgdorferi*. The vaccination with 19ISP generated humoral immunity to tick antigens (antibodies to 10 of the 19 selected antigens were detectable). The immunized guinea pigs manifested resistance to tick bites. In addition, the researchers noticed that immunization with 19ISP can protect against the transmission of *B. burgdorferi* from ticks.

In a study by Jaqueline Matias et al. [81], though they did not test the ability of *B. burgdorferi* to infect after immunization, they evaluated a strategy involving (as an antigen in the mRNA vaccine) a tick saliva protein, Salp14. It was noted that the Salp14 mRNA vaccine is capable of providing humoral immunity [81].

The formation of a pronounced antibody response is a key factor for preventing infection by *B. burgdorferi*. mRNA vaccines are a promising modality for this purpose.

### 2.3. Bordetella pertussis

*B. pertussis* is a Gram-negative *bacterium* that is spread by respiratory droplets from person to person [81]. *B. pertussis* is the causative agent of whooping cough; despite high vaccination rates among children and adults, tens of thousands of cases are still registered annually [82,83,84].

#### 2.3.1. Features of the Structure and Immune Evasion

*B. pertussis* is an obligate pathogen of humans; the bacterium produces several proteinaceous factors responsible for the symptoms observed during the disease. These include several toxins, such as dermonecrotic toxin, pertussis toxin (PT), adenylate cyclase (ACT), and Repeats in ToXins (RTX) toxin, which damage ciliated epithelial cells and alveolar macrophages and induce hyperlymphocytosis. The bacterium also produces several adhesins such as filamentous hemagglutinin (FHA), pertactin (Prn), and two fimbrial proteins (FIM2 and FIM3). *B. pertussis* has all the tools necessary to attach to host cells, to evade host defense systems, and to damage the respiratory tract [85,86]. The biosynthesis of most of the virulence factors is coordinately regulated by a two-component signal transduction system; the system consists of regulator BvgA and sensor protein BvgS [87]. Most virulence factors are secreted into the culture supernatant or displayed on the bacterial cell surface, with the exception of dermonecrotic toxin [86]. Macrolides are the standard treatment for the disease, but several strains are resistant to them [88].

#### 2.3.2. Non-mRNA Vaccines Against *B. pertussis*

There are two types of certified vaccines against pertussis: whole-cell vaccines (DTP), which are based on killed *B. pertussis* cells, and acellular vaccines (DTaP). Reactogenicity of the whole-cell vaccines has led to their replacement with acellular vaccines (DTaP), which contain one or more pertussis antigens: PT, FHA, PRN, FIM2, and/or FIM3. Acellular vaccines are able to provide protection comparable to that of whole-cell vaccines but have fewer local and systemic adverse effects [89,90,91,92,93,94]. Unfortunately, after more than a decade of use of the vaccines, there is an increase in the incidence of pertussis in industrialized countries [95,96]. Modern acellular vaccines induce an immune response of the Th2 type, whereas vaccination with bacterial cells and infection by *B. pertussis* induce an immune response of the Th1/Th17 type [97,98,99,100]. The humoral response induced by antipertussis acellular vaccines rapidly wanes, and vaccinated individuals may become asymptomatic carriers and serve as a reservoir of *B. pertussis* [100,101,102].

#### 2.3.3. mRNA Vaccines Against *B. pertussis*

An effective vaccine against pertussis should induce adaptive and protective immunity, as observed after vaccination with DTP [51]. An antipertussis multivalent mRNA vaccine has been created that also contains tetanus and diphtheria toxoid sequences [50]. It should be pointed out that all known pertussis vaccines are formulated in combination with the antigens of other pathogens, most often with diphtheria toxoid (DT) and tetanus toxoid (TT).

One such mRNA vaccine includes individual mRNA molecules encoding pertussis antigens: PTX-S1 (a truncated sequence from subunit S1 of pertussis toxoid), FHA3 (the C-terminal domain of filamentous hemagglutinin), FimD/Fim2/Fim3 (a fusion of several segments of proteins from different fimbrial serotypes), PRN, RTX (a repeat domain in toxin ACT), TCFA (tracheal cytotoxin factor A), SPHB1 (an autotransporter of subtilisin-like serine protease), and BRKA (a complement activation–inhibitory protein), as well as two mRNA molecules encoding the DT and TT antigen sequences [50]. The mRNA vaccine in that work was delivered via LNPs. The vaccine was formulated via the addition of each of the 10 mRNAs in equal amounts by weight. Mice immunized with the mRNA vaccine demonstrated humoral immunity and significantly reduced bacterial load after a challenge with two *B. pertussis* strains (UT25 and D420). The mRNA vaccine containing sequences of 10 antigens was proven to provide protection comparable to that of a recombinant-protein vaccine and whole-cell pertussis vaccine. Immunization with the mRNA vaccine suppressed leukocytosis/lymphocytosis after a challenge by *B. pertussis*, suggesting that antibodies to the pertussis toxin are able to neutralize it and prevent its negative impact on the body [50].

In another study, in Sprague Dawley rats [51], a multivalent mRNA pertussis vaccine was assessed too, which also contained sequences of subunits PTxA, PRN, FHA, FIM 2, FIM 3, BrkA, RTX, SphB1, and TCFA and two more mRNAs that encode tetanus toxoid and diphtheria toxoid. The mRNA vaccine generated humoral immunity in the rats (antigen-specific IgG antibodies). Additionally, the mRNA vaccine caused a >99% reduction in the bacterial load in the lungs, and the same trend was observed in the trachea (a 93.5% reduction in bacterial load). This evidence indicates that the mRNA–DTP vaccine results in an adaptive response and provides protection against an aerosol challenge by *B. pertussis* in Sprague Dawley rats. Nonetheless, no studies have proved the ability of mRNA vaccines to induce a protective immune response against *Corynebacterium diphtheriae* and *Clostridium tetani*, which are the source of tetanus and diphtheria toxoids. Such research would allow us to consider the clinical use of polyvalent mRNA vaccines, thereby opening up broad prospects for their simultaneous application as universal vaccines against a combination of pathogens.

### 2.4. Streptococcus pyogenes

*S. pyogenes,* which is among group A streptococci, is a Gram-positive coccus-shaped bacterium; its cell division occurs along a single axis, so that the cells can form pairs or chains while growing. Group A streptococci are regarded as one of the 10 leading infectious causes of death in humans [103,104]. *S. pyogenes* causes a variety of diseases, ranging from mild superficial infections to life-threatening illnesses with high morbidity and mortality in humans [105]; group A streptococci are the etiology of more than 288 million cases of pharyngitis annually worldwide [106]. Recurrent infections can lead to invasive dissemination of such conditions as necrotizing fasciitis, streptococcal toxic shock syndrome, and other postinfectious immune-system-related diseases [105]. Acute rheumatic fever, induced by an autoimmune response after infection with group A streptococci, is a major etiology of rheumatic heart disease, which shows high mortality rates in non-industrialized countries [107].

#### 2.4.1. Features of the Structure and Immune Evasion

Group A streptococci are obligate human pathogens, spreading primarily via airborne droplets [105,108]. Approximately 50% of the cell wall of group A streptococci is group A carbohydrate (GAC), which includes a poly(rhamnose) backbone with an N-acetylglucosamine (GlcNAc) side chain. Group A carbohydrate is highly conserved and is expressed in all *S. pyogenes* isolates [108]. Several components of group A streptococci, including the hyaluronic-acid capsule, fimbrial structures or pili, M proteins, *S. pyogenes* adhesion and division protein (SpyAD), and *S. pyogenes* fibronectin-binding protein (SfbI) (adhesin), promote the pathogen’s adhesion and colonization in the nasopharyngeal region, including the tonsillar epithelium and skin [105,109]. Group A streptococci are reported to be able to survive in phagocytic cells, and the pathogens can escape from phagolysosomes and replicate in the cytoplasm [110,111]. These pathogens’ survival is linked with streptolysin O: a secreted pore-forming toxin that promotes the release of group A streptococci from vacuoles into the cytosol of macrophages [112]. The hyaluronic-acid capsule also plays an important role in the resistance to phagocytosis, and group A streptococcal peptidase C5a—an enzyme expressed on the cell membrane—mediates resistance to phagocytosis by cleaving chemotaxin C5a on leukocytes [108,113,114]. The M protein takes part in the evasion of phagocytosis [115]. The evasion of immune cells is mediated by protease SpyCEP through cleavage of IL-8 [116]. Group A streptococci are capable of producing exotoxins that act as superantigens. These superantigens are responsible for the clinical features and high mortality rate of streptococcal toxic shock syndrome [117]. Superantigens are immunomodulatory proteins that stimulate T cells by directly binding to class II molecules. Superantigens bypass antigen processing and major histocompatibility complex (MHC) restriction, thereby inducing T-cell stimulation and a release of proinflammatory cytokines [118]. This intense inflammatory cascade is responsible for the clinical features of streptococcal toxic shock syndrome [119].

#### 2.4.2. Non-mRNA Vaccines Against *S. pyogenes*

The treatment of infections caused by group A streptococci is mostly based on antibiotics. Humans are the only known natural reservoir of group A streptococci, and therefore a safe and effective vaccine should reduce the burden on human health and may completely eliminate the pathogen. All vaccines against group A streptococci can be categorized into two types: vaccines based on the M protein (encoded by the *emm* gene) and vaccines based on a non-M protein. The second type of vaccine targets various proteinaceous factors of group A streptococci, such as pyrogenic exotoxins, *S. pyogenes* adhesion and division protein (SpyAD), *S. pyogenes* cell wall proteinase (SpyCEP), streptolysin O, and peptidase C5a and carbohydrate-based antigens [120,121].

The first type of vaccines includes a 30-valent vaccine called StreptAnova, which targets N-terminal epitopes of the M protein. The vaccine passed phase I clinical trials in 2020, which showed the safety of the vaccine and its ability to generate an adaptive immune response without inducing autoimmunity or cross-reactive antibodies [122]. Another vaccine based on the M protein is 26-valent vaccine StreptAvax. This vaccine has induced strong protection from streptococcal infections [123]. Nevertheless, its development has been discontinued for commercial reasons [122]. An additional domain of the M protein from the C-terminal region has been investigated as a vaccine candidate: J8 (MJ8VAX). A double-blind randomized phase I clinical trial indicates that MJ8VAX is safe and provides an adaptive immune response, but antibody levels decline over time. The efficacy of the vaccine is unstable judging by the data from a small number of participants [124]. Vaccines J8/S2 combivax and P*17/S2 combivax contain K4S2 (a B-cell epitope-modified peptide from *S. pyogenes* cell coat proteinase [SpyCEP]) in combination with a peptide from the C-terminus of the M protein: J8 for J8/S2 combivax and P*17 for P*17/S2 [125,126]. In a mouse study, vaccines J8/S2 combivax and P*17/S2 combivax protected against cutaneous and systemic infection with hypervirulent CovR/S strains of *S. pyogenes* [125]. The developers have received approval for phase I trials of vaccines J8/S2 combivax and P*17/S2 combivax [121,127]. StreptInCor is composed of a 55-amino-acid peptide from the M protein’s conserved regions featuring B- and T-cell epitopes [128]. In preclinical studies, StreptInCor has induced high levels of antigen-specific antibodies and good survival vis-a-vis *S. pyogenes* infection in mice and has shown the absence of autoimmune reactions [128,129]. In minipigs, the vaccination has been well tolerated and has not had an adverse effect on cardiac tissue [130].

Vaccines based on a non-M protein are also being devised, but none of which are currently in clinical trials. Vaccine Combo4 contains the native group A carbohydrate of *S. pyogenes* conjugated to carrier protein CRM 197 as well as SLO, SpyCEP, and SpyAD [131]. Preclinical studies on Combo4 have revealed immune protection in mouse models and efficacy in opsonophagocytic-killing assays based on immunized-rabbit serum samples [132,133]. VAX-A1 produced by Vaxcyte contains a modified version of group A carbohydrate in which the GlcNAc side chain has been removed. Immunization of mice with the VAX-A1 vaccine has protected against a challenge with *S. pyogenes* in both a systemic-infection model and a localized cutaneous-infection model [134]. The candidate vaccine Combo5 has protected mice against superficial skin infections but not against invasive disease [135]. The addition of adjuvant SMQ to Combo5 provides protection against an invasive challenge in mice, possibly owing to a more balanced Th1/Th2 immune response compared to Combo5 containing alum as an adjuvant; the latter vaccine induces a Th2-biased response [135]. Notably, in the article just cited, Combo5/SMQ protected mice from the invasive challenge in the absence of opsonizing antibodies, suggesting that opsonizing-antibody responses may not necessarily correlate with protection for M protein-free vaccines [135]. TeeVax is a multivalent vaccine targeting T antigens, the main protein component of *S. pyogenes* pili, which are surface-exposed and involved in host adhesion and colonization during infection [136]. TeeVax consists of three recombinant proteins, each being a fusion of six unique T-antigen domains. The combination of all three proteins elicits a sustainable antibody response in rabbits that is reactive with all 18 T antigens [137].

#### 2.4.3. mRNA Vaccines Against *S. pyogenes*

There are several mRNA vaccines against *S. pyogenes* that are in preclinical development. In one paper, a vaccine based on saRNA was designed [138]. In that work, the efficacy of saRNA vaccines containing mutant streptolysin O (SLOdm) sequences was assessed (SLOdm plays a key role in the pathogenesis and can induce protective antibodies in mouse models of the infection [138]). The vaccine was delivered using cationic nanoemulsion. The latter is a nonviral delivery system based on the proprietary adjuvant Novartis MF59, which is safe and well tolerated in children, adults, and the elderly [139]. The vaccine also induces substantial amounts of functional serum antibodies and protects mice from bacterial infection [52].

A recent study assessed the immunogenicity of a 30-valent mRNA vaccine that has been developed based on the corresponding protein vaccines. A 30-valent M protein–based vaccine has previously been found to induce an adaptive response in animal models and in a phase I clinical trial [122]. The mRNA vaccine in question encodes 30 N-terminal M peptides with a C-terminal transmembrane anchor (CtTM). The vaccine was delivered using LNPs. The 30-valent mRNA–LNP Strep A vaccine induced adaptive immunity and opsonophagocytic antibody production similar to the protein vaccine. The mRNA platform has good potential to encode more N-terminal M peptides than the protein version can carry.

Thus, the mRNA vaccines developed to date are capable of eliciting functional antibodies against group A streptococci, thereby inducing protective immunity against invasive infections in animals. The RNA vaccine platform represents a promising tool in the fight against group A streptococci.

### 2.5. P. aeruginosa

*P. aeruginosa* is a Gram-negative rod-shaped bacterium, an opportunistic pathogen found ubiquitously in the environment. *P. aeruginosa* is the etiological factor of leading nosocomial infections [140]. It can cause an acute or chronic infection in individuals at risk, including people with weakened immunity, neutropenia, chronic obstructive pulmonary disease, mucoviscidosis, burns, or various implants [140,141,142,143].

#### 2.5.1. Features of the Structure and Immune Evasion

On the surface of *P. aeruginosa*, there is a single flagellum and type IV pili, which provide mobility of the bacterium and its fixation on the surface of epithelial cells owing to binding to mucins and glycosphingolipids (asialo-GM1 and asialo-GM2) [144,145]. In *P. aeruginosa*, six types of secretion systems (T1SS to T6SS) have been identified that are capable of transporting various bacterial biomolecules to the outer membrane of the bacterium, into the extracellular environment, or directly into a host cell [146]. It has been reported that the presence of the type III secretion system—which, along with the T6SS, is capable of introducing secreted substances into a host cell—contributes to enhanced virulence of *P. aeruginosa*, higher bacterial load, and higher mortality [144]. Secreted effector proteins and toxins are involved in the regulation of such processes as host cell apoptosis, modulation of the host’s immune response, colonization, biofilm formation, quorum sensing, bacterial interactions, and competition with microorganisms in the environment [147].

The hypermutability of *P. aeruginosa* is its main feature relevant to the development of resistance mechanisms against a wide range of antibiotics. The most investigated and widespread mechanisms include the following: (1) A decrease in the number of nonspecific porins and their replacement with more specific ones, which leads to considerable restriction of antibiotic intake. (2) The presence of active efflux pumps and their hyperproduction—*P. aeruginosa* has four of the best-known efflux pumps (MexAB-OprM, MexXY/OprM(OprA), MexCD-OprJ, and MexEF-OprN), which are capable of removing many antimicrobial drugs from the cell. (3) AmpC hyperproduction, which occurs mainly due to mutations in *ampD* (repressor of *ampC*) and point mutations in *ampR*, which are involved in the induction of *ampC*. This is the reason for the resistance of *P. aeruginosa* to most β-lactams, monobactams, and third- and fourth-generation cephalosporins. (4) The regulation of genes by the quorum sensing system, participating in the development of antibiotic resistance mechanisms. (5) The emergence of an extracellular polymer matrix during biofilm formation; this matrix protects the bacterium from the environment, including some antibiotics. Finally, of course, (6) there is horizontal gene transfer [140,147,148]. Thus, the antibiotic resistance of *P. aeruginosa* is a global problem, and the design of new antimicrobial therapy strategies has many limitations [149].

#### 2.5.2. Non-mRNA Vaccines Against *P. aeruginosa*

Despite the urgent need for prophylactic measures to prevent infection, no vaccine against *P. aeruginosa* has yet been approved. To date, only three candidate vaccines have reached phase III clinical trials: (1) a bivalent vaccine containing antigens from several flagellum subtypes [150]; (2) Aerugen^®^, a conjugate vaccine consisting of exotoxin A as well as O-polysaccharide from eight serotypes of *P. aeruginosa* [151]; and (3) IC43, a recombinant vaccine in which there are fusion proteins of epitopes of OprF and OprI [152]. Nonetheless, clinical trials of the created vaccines have not been successful.

#### 2.5.3. mRNA Vaccines Against *P. aeruginosa*

Several studies on mRNA vaccines against *P. aeruginosa* are known (Table 2). The first work on the role of mRNA vaccines against *P. aeruginosa* was recently conducted by Xingyun Wang et al. [54]. In that paper, two variants of mRNA vaccines were investigated: (a) with PcrV antigen (a component of the TSS3 of *P. aeruginosa*), and (b) with fusion protein OprF-I (composed of the outer-membrane proteins OprF and OprI).

Mice have been immunized with PcrV and OprF-I mRNA vaccines either separately or in combination. PcrV and OprF are well-studied, highly conserved, and virulent antigens among all strains of *P. aeruginosa* and have earlier been used in the protein recombinant vaccine IC43. It has been demonstrated that the mRNA vaccine type involving the sequence of the PcrV antigen gives rise to a more pronounced adaptive immune response compared to OprF-I [54]. Immunization with both mRNA vaccine types induces high titers of IgG, good survival of mice, a decrease in the bacterial load of organs, and the attenuation of inflammatory changes and damage to lung tissue after the infection compared to a control group of animals without the administration of the vaccine.

Nevertheless, in that work, the number of splenocytes secreting IFN-γ was significantly higher in mice immunized with the mRNA vaccine encoding the PcrV antigen than in mice immunized with the vaccine coding for the OprF-I antigen. Further, intramuscular vaccination with mRNA–PcrV was found to mostly induce CD4^+^ T-cell responses. After immunization with both vaccines, a Th1/Th2 or slightly Th1-biased immune response was observed.

The efficacy of PcrV as an antigen in an mRNA vaccine is supported by another study from Kawaguchi et al. [153]. Two mRNA vaccines with the PcrV antigen sequence were tested there, one with (ss-pcrV-f) and one without (ss-pcrV) the trimerization domain. The titer of IgG antibodies specific to PcrV was significantly higher with both vaccines compared to unvaccinated animals. Furthermore, after the intratracheal infection of mice with strain PA103, the titer of IgA specific to PcrV was measured in the lungs; this antibody class plays an important part in mucosal immunity. Immunization with the mRNA vaccine encoding the trimerization domain resulted in elevated titers of IgA in comparison with unvaccinated animals. Additionally, immunization with either mRNA vaccine (ss-pcrV or ss-pcrV-f) increased the survival of mice by 75.0% and 62.5%, respectively, and reduced the bacterial load in the lungs of the animals by 10–30-fold. Mice vaccinated with either ss-pcrV or ss-pcrV-f showed lower myeloperoxidase activity and diminished IL-6 levels in the lungs. The vaccinated mice also showed virtually no neutrophil infiltration of lung tissue, alveolar hemorrhage, or disintegration of the alveolar structure compared to the controls. Altogether, these findings indicate a reduction in lung inflammation. Thus, the use of PcrV-based mRNA vaccines has caused an adaptive and protective response against *P. aeruginosa* [154].

### 2.6. Listeria monocytogenes

*L. monocytogenes* is a Gram-positive bacterium that causes a disease called listeriosis. In the 1980s, it was identified as a foodborne pathogen [155]. *L. monocytogenes* is quite tolerant of environmental stressors, low pH, and high ionic strength, and is capable of growing at low temperatures; these characteristics make *L. monocytogenes* a serious problem for the food industry [156]. After ingestion of contaminated food, most people experience gastroenteritis of various severity levels [155,157]. *L. monocytogenes* is the third leading cause of death from foodborne bacteria in the United States [158].

#### 2.6.1. Features of the Structure and Immune Evasion

Listeria, just like other pathogens, can quickly develop resistance to antimicrobial agents, thereby posing a threat to human and animal health. Isolates of *L. monocytogenes* have shown various degrees of resistance to commonly used antibiotics, and the first multidrug-resistant isolate of *L. monocytogenes* was identified in France in 1988 [159]. There is also a report of infections caused by the advent of antibiotic-resistant *Listeria* strains isolated from foods [160].

After ingestion, *L. monocytogenes* enters the intestine, penetrates the epithelial barrier, and then spreads through lymph and blood to target organs: the liver and spleen [161,162]. Once in the bloodstream, *L. monocytogenes* can penetrate the blood–brain barrier, thus causing central-nervous-system infections [163], or may cross the fetoplacental barrier, thereby inducing miscarriage or pregnancy complications [164].

A distinctive feature of *L. monocytogenes* is its ability to enter nonphagocytic cells via receptor-mediated endocytosis involving its surface-expressed virulence factors internalin (Inl) A and InlB [165,166]. These two interact with cellular receptors E-cadherin and Met, and this event leads to bacterial engulfment [167]. Once inside an endocytic vesicle, *L. monocytogenes* secretes phospholipase (Plc) A (also known as 1-phosphatidyl inositol phosphodiesterase), PlcB, and pore-forming toxin listeriolysin O (LLO); the latter helps the bacterium disrupt the endosome membrane and enter the cell cytoplasm [168]. Another feature of the intracellular lifestyle of *L. monocytogenes* is the capacity to polymerize actin and spread from cell to cell using a surface protein, ActA. The protein induces actin assembly at a bacterial-cell pole—via interaction with the ARP2/3 complex, which mediates actin polymerization—and ensures the cell-to-cell spread of the bacterium [169,170,171].

With the intracellular lifestyle, the main strategy of a host’s defense against *L. monocytogenes* is CD8^+^ T-cell-mediated cytotoxicity; accordingly, an effective vaccine should be able to induce cellular immunity [172]. *L. monocytogenes* antigens are efficiently presented by MHC-I and MHC-II, thereby stimulating both CD4^+^ and CD8^+^ T-cell immunity [173]. Activated T lymphocytes generate an adaptive immune response by inducing the production of Th1 cytokines IFN-γ, TNF, and IL-12 [174,175]. Activated CD8^+^ CTLs secrete perforin, granzyme, granulysin, and other proteins, thus killing infected cells along with their intracellular *L. monocytogenes* [176,177].

#### 2.6.2. Non-mRNA Vaccines Against *L. monocytogenes*

To date, live attenuated vaccines against *L. monocytogenes* have been researched well. Immunization with mutant strains induces CD8^+^ T-cell responses and provides protective immunity against *L. monocytogenes* in animals [178,179,180,181,182,183,184]. On the other hand, attenuated vaccines show genetic instability during long periods of time, thereby posing the risk that the attenuated strain will revert to virulent status [185,186]. Aside from attenuated strains, cell-based, DNA, viral-vector, subunit, and recombinant-protein vaccines have been evaluated against listeriosis [187,188,189,190,191]. Antigens of *L. monocytogenes* that contribute to protective immunity include predominantly the non-spore-forming variant of listeriolysin O (LLO) and invasion-associated protein p60 (p60/iap), as well as glyceraldehyde-3-phosphate dehydrogenase (GAPDH) [192]. The use of live vaccines in people with any type of immunosuppression poses serious difficulties; subunit vaccines are safer but require adjuvants to enhance immune response, and these adjuvants can exert adverse effects [193].

#### 2.6.3. mRNA Vaccines Against *L. monocytogenes*

mRNA vaccines against listeriosis have been tested in animal models [55]. In one study, for the delivery of mRNA, the researchers formulated it into cationic LNPs; glycolipid α-GC served as an adjuvant (this mRNA delivery platform promotes pluripotent innate and adaptive immune responses [194]). mRNA encoding protein LMON_0149 (ABC oligopeptide transporter and oligopeptide-binding periplasmic protein OppA) significantly reduced the bacterial load in the livers of the vaccinated animals. Vaccination with the mRNA vaccine resulted in a 1000-fold reduction in bacterial load in the spleen compared to the control animals. The mRNA vaccine generated a T-cell response, which is necessary for protection from *Listeria* [172]. It has also been reported that a combination of two mRNA vaccines containing sequences of antigen LLO_E262K (a non-pore-forming variant of listeriolysin O) and LMON_2272 (protein OppA in *Listeria*) produces a significant reduction in bacterial load within both the spleen and liver, similarly to LMON_0149 [55].

Other research articles [195,196] show the ability of mRNA-encoding bacterial antigen GAPDH (glyceraldehyde-3-phosphate dehydrogenase) from *L. monocytogenes* to activate dendritic cells. Protein GAPDH isolated from *Listeria* shares more than 95% homology with the N-terminal peptide of GAPDH from *Mycobacterium marinum* or *Streptococcus pneumoniae*. Vaccinated mice turned out to be protected not only from infection with *L. monocytogenes* but also from *M. marinum* or *S. pneumoniae* [195,196].

Consequently, mRNA vaccines are capable of providing protection to animals against the intracellular pathogen *L. monocytogenes*. The mRNA platform allows us to encode a large number of epitopes, which will potentially improve protection from listeriosis in humans and animals.

### 2.7. Chlamydia trachomatis

*C. trachomatis* is a Gram-negative, obligate intracellular bacterium with a unique life cycle that infects the epithelium of the eye and genital tracts. As an intracellular pathogen, *C. trachomatis* reproduces within the membrane vacuole of host cells, and this arrangement allows it to evade immune responses and survive inside the cell [197,198].

#### 2.7.1. Features of the Structure and Immune Evasion

*C. trachomatis* is primarily transmitted sexually. The pathogen is widespread throughout the world and is a substantial public health problem [198]. In 80% of cases in females and 50% of cases in males, the infection is asymptomatic, which contributes to its widespread prevalence among humans and delayed treatment [197,199]. *C. trachomatis* can cause pelvic inflammatory disease, ectopic pregnancy, and infertility due to tubal damage [200,201]. With respect to ocular infections by the conjunctivitis strains of *C. trachomatis*, the pathogen can induce conjunctivitis and trachoma [202]. Furthermore, the development of invasive conjunctivitis in adults can be the only manifested symptom of the sexually transmitted infection [203,204].

*C. trachomatis* uses a number of structural adaptations for survival and reproduction, and these mechanisms contribute to its pathogenicity and the chronicity of the infections. One of the key mechanisms is the activation of signaling pathways through the major outer-membrane protein (MOMP), which constitutes 60% of the membrane by weight [205]. MOMP functions as an adhesin, helping the pathogen to attach to host cells through electrostatic and hydrophobic interactions [206,207,208]. The infection cycle of *C. trachomatis* involves two forms of the pathogen. The infective form—elementary bodies (EBs)—enter the host cell and transform into reticulate bodies (RBs), which replicate using the cell’s resources. The reticulate bodies then transform back into elementary bodies, which exit the cell via lysis or extrusion, thereby continuing the cycle in neighboring cells [207,208,209].

The type III secretion system (T3SS) modulates differentiation between reticulate bodies and elementary bodies, whereas microtubules and proteins such as 14-3-3β maintain bacterial nutrition and block apoptosis, thus helping *Chlamydia* to survive within host cells [208,210,211]. Genetic variability and the plasticity zone also contribute to the capacity for adaptation to host conditions while helping to evade immune responses and supporting the survival of the bacterium during tryptophan depletion [212].

The ability of this pathogen to evade an immune response is due to a decrease in the surface expression of MHC molecules types I and II in both infected and neighboring epithelial cells [205,213]. For this reason, an immune response is underactivated, allowing the infection to persist and become chronic [213,214]. Infection by *C. trachomatis* induces an immune response characterized by the proliferation of lymphocytes specific to MOMP. The MOMP of *C. trachomatis* can activate Toll-like receptor 2 (TLR2)-signaling pathways, leading to the secretion of such inflammatory cytokines as IL-8 and IL-6. This activation may promote local immune responses and pathology and may be useful as a target for vaccine development [214].

#### 2.7.2. Non-mRNA Vaccines Against *C. trachomatis*

Despite the high prevalence of this infection worldwide, there is still no effective vaccine. Antibiotic therapy remains the mainstay of treatment of chlamydial infection. Azithromycin for uncomplicated sexually transmitted infections and doxycycline for eye infections is commonly used [200,202].

#### 2.7.3. mRNA Vaccines Against *C. trachomatis*

A recent study [56] involved a vaccine based on saRNA encapsulated into liposomes or emulsions with cationic lipids. The vaccine based on saRNA-encoding MOMP provided both humoral (judging by IgG titers) and cellular immune responses (according to ELISpot assay for IFN-γ). In particular, MOMP-specific IgG titers reached 100 ng/mL at week 9 postimmunization. Antigen expression also persisted for up to 60 days [56].

Thus, the administration of mRNA vaccines as a preventive measure against *C. trachomatis* has yielded encouraging results in mice.

### 2.8. Yersinia pestis

*Y. pestis* is a Gram-negative bacterium having a coccobacillus shape and is a facultative intracellular pathogen that can reproduce inside macrophages and effectively evade a host’s immune defenses [215]. Key features of this plague pathogen are the inhibition of phagocytosis, survival, and reproduction inside macrophages. *Y. pestis* is able to modulate a host’s immune response, thereby creating an imbalance between proinflammatory and anti-inflammatory cytokines, which increases the risk of complications [216].

#### 2.8.1. Features of the Structure and Immune Evasion

Upon contact of the bacterium with phagocytic cells of a mammalian host, the type III secretory system (T3SS) is activated, allowing Yop proteins to penetrate the cell membrane. Inside the cell, Yops suppress phagocytosis and the synthesis of proinflammatory cytokines. For example, the YopH protein blocks the activation of T and B lymphocytes, whereas YopE, YopT, and YpkA disturb the function of Rho GTPase and cytoskeleton structures [217,218]. The LcrV protein (antigen V), when released through the T3SS, stimulates the production of immunosuppressive IL-10, which suppresses inflammatory processes and facilitates the survival of the bacterium by interacting with TLR2 and CD14, resulting in a release of IL-10 [219,220]. The molecular features of *Y. pestis* virulence are closely related to the temperature-dependent expression of its pathogenic factors. At 37 °C, the most important mechanisms that ensure the survival and spread of the bacterium are activated. Under these conditions, the F1 antigen is expressed, forming a protective capsule that helps the pathogen to avoid phagocytosis [221]. The structure of lipopolysaccharides (LPS) also changes; lipid A is converted from a hexa-acylated form to a tetra-acylated one, which reduces recognition of the bacterium by receptor TLR4 and helps evade a host’s immune response [219,222]. Lipid A interacts with the TLR4–MD-2 receptor complex, thus triggering an immune response. CD14 on the cell surface enhances this process by transferring LPS to TLR4–MD-2. Adapter proteins such as MyD88 and TRIF then trigger intracellular TLR4-signaling pathways, thereby allowing the pathogen to evade an immune response [219,222]. *Y. pestis* also adapts to survival in mammals through alteration of the O antigen. The latter is absent in its LPS owing to genetic changes, and this state of affairs activates the protease Pla, which facilitates the spread of the infection. Protease Pla, which is present in the outer membrane of *Y. pestis*, activates plasminogen, degrades complement components (C3), inhibits phagocytosis, and cleaves and inactivates the tissue factor pathway inhibitor (TFPI; a key inhibitor of the coagulation cascade), thereby helping the pathogen to evade immune defenses. Additionally, adhesin Ail interacts with complement inhibitors such as C4b and vitronectin, thus blocking the complement system [223,224].

Although humoral immunity is important for protection against plague, this immunity may not be sufficient to prevent the pneumonic type of the disease. Vaccines based on proteins F1 and LcrV can protect small animals, but research suggests that antibodies may be insufficient to protect primates from a *Y. pestis* aerosol challenge, despite high antibody titers in immunized animals [219].

#### 2.8.2. Non-mRNA Vaccines Against *Y. pestis*

The development of plague vaccines is of utmost importance due to the high pathogenicity of *Y. pestis*, which causes three main types of infection: bubonic, pneumonic (lung), and septicemic. The most dangerous is the pulmonary type, which is transmitted by airborne droplets [219,222]. Of special concern are the strains of *Y. pestis* having multidrug resistance. Although such strains are rare, they considerably complicate treatment because standard antibiotics may be ineffective. This situation poses a serious threat to public health and points to the need to devise effective vaccines [219,225].

A live plague vaccine (LPV), created on the basis of strain EV76, was first successfully employed during an epidemic in Madagascar in 1930. Despite its ease of production, LPV provides only short-term protection, requiring annual revaccination [226,227]. A possible reason is that the key antigen LcrV elicits only a weak immune response [227,228].

#### 2.8.3. mRNA Vaccines Against *Y. pestis*

Research in the field of mRNA vaccines against *Y. pestis* is aimed at creating modalities encoding the key antigens F1 and LcrV. In one of the recent projects [58], a single-dose mRNA vaccine was developed, encoding the F1 antigen bound to human signaling peptide Fc. It manifested 100% efficacy by providing complete protective immunity in mice against virulent *Y. pestis* strain KIM53 (100LD50) after a single administration [58]. In another study [57], a vaccine based on saRNA was designed that encodes two key antigens: F1 and V. Experiments revealed that it effectively protects against a virulent clinical isolate of *Y. pestis* from Madagascar. The vaccine stimulated the production of specific antibodies against antigens F1 and LcrV, thereby providing protection from both encapsulated and acapsular plague strains [57]. The saRNA vaccine effectively induced humoral and cellular immune responses, as confirmed by high levels of antibodies to antigens F1 and LcrV. The vaccination ensured protective immunity against lethal infection, with most animals surviving infection by *Y. pestis* [57].

Therefore, the results of these studies indicate that the application of mRNA-based vaccines against plague may be a promising area for research and development.

### 2.9. Rhodococcus equi

*R. equi* is a Gram-positive, facultative intracellular zoonotic bacterium that can cause serious pulmonary infections primarily in animals or in immunocompromised humans [229,230].

#### 2.9.1. Features of the Structure and Immune Evasion

*R. equi* is a major etiological factor of pneumonia in foals aged 2 to 6 months owing to decreased levels of maternal antibodies [231]. In humans, *R. equi* causes pyogranulomatous bronchopneumonia, which mainly affects immunocompromised patients, e.g., HIV-infected people [232].

When *R. equi* enters the body, the pathogen is engulfed by alveolar macrophages, but virulent strains are able to evade phagosome death and reproduce within the cells, thereby leading to pyogranulomatous pneumonia [231]. Pathogenicity of *R. equi* is linked to the presence of virulent plasmids of the pVap type, which contain genes that ensure survival and the ability to reproduce inside macrophages [231]. Plasmid pVapA performs a key function in the virulence of strains that infect horses and foals [229]. The VapA protein, expressed on the surface of this bacterium, is accessible to the immune system and promotes the survival of the bacterium in phagosomes by preventing their acidification and disrupting cellular membranes, thus creating conditions for the pathogen’s reproduction [233]. Infections caused by *R. equi* differ between animals and humans; only 20–25% of human isolates express VapA [234].

Transfusion of hyperimmune plasma and prophylaxis with macrolides and rifampicin are used to prevent and treat the infection in foals. Not so long ago, however, the acquisition of resistance by the pathogen toward the main therapeutic antibiotics—macrolides and rifampicin—raised some concerns [232]. The resistance is reported to be associated with mutations in the 23S rRNA gene (macrolide resistance) and in the *rpoB* gene (rifampicin resistance), thereby complicating the treatment for foals [232]. Moreover, methyltransferase genes *erm(46)* and *erm(51)*—responsible for antibiotic resistance—are spread through horizontal gene transfer, thus posing a threat to other bacterial species in the environment [229].

#### 2.9.2. mRNA Vaccines Against *R. equi*

There is currently no approved effective vaccine for the prevention of *R. equi* infection. A promising avenue of research is the use of mRNA technologies to create a vaccine against *R. equi*. Lately [59], against *R. equi*, investigators have utilized an mRNA vaccine that is capable of inducing both systemic and local immune responses; of the four mRNA vaccine versions encoding different variants of the VapA protein, construct mod-VapA featuring deletion of the transmembrane domain was chosen in that work [59]. This mRNA vaccine showed the greatest efficacy in the vaccination of foals.

In that study, humoral and cellular immune responses were assessed in newborn foals after administration of a nebulized (sprayed) or intramuscular mRNA–LNP vaccine coding for the VapA protein. The intramuscular vaccination proved to be more effective in stimulating a systemic immune response, whereas the nebulized vaccine stimulated local immunity of the respiratory tract [9,59]. These data highlight good prospects for further development of mRNA vaccines against *R. equi* taking into account different routes of administration to optimize both local and systemic immunity.

## 3. In Silico Approach

Immunobioinformatic methods are currently widely used to create mRNA vaccines [9]. Rapid advances in bioinformatic and artificial intelligence techniques, including machine learning and neural networks, have enabled researchers to create many tools for the analysis and prediction of antigen presentation [235,236]. Currently, the most promising strategy for choosing an antigen for vaccines is the design of a multiepitope antigen that consists of many individual epitopes connected by linkers [9]. Certain components of full-length proteins can have undesirable immune consequences or simply do not have the necessary activities, thereby reducing the effectiveness of the vaccine. A multiepitope vaccine allows one to select the most immunogenic, safe (i.e., not causing an allergic, autoimmune, or cytotoxic response), and most frequently occurring epitopes in a population. In addition to epitope selection, in silico approaches help an investigator to determine the secondary and tertiary structure of a polypeptide of interest, to conduct molecular docking, and even to predict potential dynamics of immune responses and immunity, thereby allowing for the selection of an optimal vaccine administration regimen [9]. Nonetheless, most of the in silico studies on the design of vaccines against bacterial pathogens remain without experimental confirmation of effectiveness in animal models [237,238,239,240]. Evidently, several more years or even decades of comprehensive research (in silico prediction/experimental confirmation)—with the constant progressive optimization of algorithms depending on the obtained data—are needed to obtain an effective vaccine.

An ideal multiepitope vaccine should be designed to include epitopes that can elicit CTL and helper T lymphocyte responses and a B-cell response [9,241]. The basic principles of antigen selection mean selecting the most common pathogenic strain and searching for candidate proteins from whose epitopes the antigen is formed. Typically, such proteins are assumed to be surface proteins of the pathogen, which primarily come into contact with components of the immune system. Of note, there are limitations related to the absence of an annotated proteome of some bacterial pathogens.

Five basic sets of procedures can be highlighted that are necessary for devising a multiepitope vaccine for mass use. The first one is a search for immunogenic HTL, CTL, and B-cell epitopes among known protein sequences of an infectious agent in question. For this purpose, the Immune Epitope Database Analysis Resource (IEDB-AR) is used, which contains experimental data on the identification and characterization of epitopes and epitope-specific immune receptors [242]. Next, these epitopes are characterized for antigenicity, allergenicity, and toxicity, and for this purpose, the servers Vaxijen, AllerTOP, and Toxinpred are most often employed [243,244,245]. Antigenicity, allergenicity, and toxicity are assessed via a comparison with known antigenic/allergenic and toxic epitopes, respectively, after auto cross-covariance transformation of protein sequences into uniform equal-length vectors. This transformation allows one to take advantage of an alignment-free approach based on the main physicochemical properties of protein sequences, e.g., amino acid hydrophobicity, molecular size, helix-forming propensity, relative abundance of amino acids, and β-strand-forming propensity. The third set of analyses includes assessments of the ternary structure of the protein and disulfide bonds, molecular docking with target receptors of lymphocytes, and molecular dynamics stimulation. The final two steps include the evaluation of the frequency of the specific HLA1 and HLA2 complexes (in a population) that show the strongest affinity for an epitope of interest as well as in silico cloning and evaluating the possibility of codon optimization to boost the efficiency of production in model microbes such as *Escherichia coli* or *Saccharomyces cerevisiae*. Although in silico tools enable us to predict the efficiency of future vaccines with a high degree of accuracy, in vivo studies are still necessary to understand their efficacy and safety.

Aside from the analysis of the epitopes themselves, (i) sequences of linkers that will connect them and (ii) the presence or absence of adjuvant sequences or localization signal sequences play an important role in the efficiency of a vaccine. Linkers AAY, GPGPG, and KK are widely used to join CTL, HTL, and B-cell epitopes, respectively, although some researchers utilize other linker sequences such as HEYGAEALERAG, GGGS, and RVRR [246,247]. There are many more variations in terms of adjuvants [248], and each of them has biological activities. Nonetheless, the number of comprehensive studies evaluating different combinations of linkers or adjuvants for multiepitope vaccines is currently inadequate.

To the best of our knowledge, there are several avenues of research for improving multiepitope vaccines. First, because multiepitope vaccines are developed for a specific pathogen, one promising field in this area is the creation of cross-reactive vaccines that recognize common molecular patterns in pathogens [196]. The second field is the optimization of linker and adjuvant sequences. The third field is the inclusion (in a vaccine) of sequences encoding epitopes for “unconventional” MHC molecules such as MHC-E; this approach may increase the efficacy of the obtained vaccine [249].

## 4. Discussion

Existing studies on the use of mRNA vaccines against bacterial infections indicate that these vaccines in most cases provide humoral and cellular immunity. Some research articles have also shown the formation of protective immunity that reduces the bacterial load and improves the survival of experimental animals after a challenge with a pathogenic bacterium. Nevertheless, it should be recognized that the development of mRNA vaccines against bacterial pathogens is at a rudimentary stage. Only three candidate vaccines are in clinical trials; the other vaccines have been tested only on rodent models.

Although trial results have not yet been presented, the protective effects obtained in animal models are expected to be similar in humans. The specific features of the immune response of animal models to some bacterial infections are mitigated by selecting a more appropriate model, such as the use of guinea pigs instead of mice, which are the natural intermediate hosts of ticks that carry *B. burgdorferi,* when studying the mRNA vaccine against *B. pertussis* with tick saliva antigens.

The mRNA therapeutics platform has recently been actively studied and optimized by researchers. Although the mRNA vaccine against bacterial infections is a relatively new and poorly understood area, the advances made in modifying the mRNA vaccine as a platform are also applicable to bacterial infections. Improvements that affect the effectiveness of mRNA therapeutics as a platform include modified nucleotides, cap analogs, the rational selection of UTRs, codon optimization, secondary structure analysis, and improved mRNA therapeutics delivery [250,251,252,253,254,255]. These modifications are universal and applicable to the mRNA vaccine regardless of the antigen sequence. However, in the context of mRNA vaccines against bacterial infections, the question of the strategy for optimizations aimed at the correct synthesis of bacterial proteins in a eukaryotic cell remains open. For the successful use of mRNA vaccines against bacterial infections, a large amount of research has yet to be conducted regarding the choice of an optimal antigen sequence and signal and adjuvant sequences for mRNA vaccines. Advancements in immunoinformatic methods should help speed up these processes, but as mentioned above, without experimental testing on various species of animals (not only rodents but also nonhuman primates), it is impossible to obtain effective modalities for clinical use in humans.

Some bacteria are able to evade the innate response, some the adaptive response, and some have adapted to evade both. Accordingly, it is necessary to consider improving the response of not only the adaptive immune response but also the innate response through vaccination or other methods. It is known that some infections can be effectively suppressed by innate immunity in the early stages of infection [256].

At present, there are no studies on the therapeutic effect of mRNA vaccines against bacterial infections, but there are studies where therapeutic immunization with other types of vaccines in combination with antibiotics provides a better treatment effect than antibiotic therapy alone [257]; the therapeutic effect of mRNA vaccines against bacterial infections in combination with antibiotics should be studied in more detail.

## 5. Conclusions

In summary, the development of mRNA vaccines against bacterial diseases is well underway. There is no doubt that the first mRNA-based vaccines will be approved and used in humans in the next few years. Of particular importance is the development of mRNA vaccines against *M. tuberculosis*, *B. burgdorferi*, *B. pertussis*, and *S. pyogenes*, as the diseases caused by these pathogens are of major social importance, causing a high number of deaths and a significant burden on healthcare systems. In addition, the availability of already licensed vaccines (not based on mRNA) against these pathogens indicates the effectiveness of the vaccination strategy in the fight against these pathogens. Among other bacterial pathogens, *K. pneumoniae* and *S. aureus* were not included in this review, but the development of mRNA vaccines against these pathogens is an urgent task.

On the basis of the evidence from existing studies, several promising areas for improvements in mRNA vaccines against bacterial infections can be underscored. The first area is the use of multiepitope and multivalent vaccines or a combination of these approaches (i.e., when epitope sequences of different bacterial pathogens are encoded in one RNA molecule). This approach may help to create universally effective and, most importantly, the safest mRNA vaccines against several bacterial pathogens. The second area is the administration of mRNA vaccines via various heterologous prime–boost schemes along with existing non-mRNA vaccines. Third, taking advantage of new alternative platforms such as saRNA and/or circular RNAs also seems to be a promising field for the development of mRNA vaccines against bacterial pathogens. 

## Figures and Tables

**Table 1 ijms-25-13139-t001:** The list of bacteria against which RNA vaccines are being developed.

Bacterium	Bacterium Type	Statistics	RNA Vaccine Development Stage	Approved Vaccines
*Mycobacterium* *tuberculosis*	Gram-positive	Cases: 10.6 million ^1^ Deaths: 1.6 million ^1^	phase I (NCT05547464)	BCG
*Borrelia* *burgdorferi*	Gram-negative	Cases: 476,000 ^2^ Deaths: n.a.	phase I (NCT05975099)	Phase III (VLA15) (NCT05477524)
*Bordetella* *pertussis*	Gram-negative	Cases: 101,500 ^1^ Deaths: n.a.	In vivo animal studies	DTP (diphtheria-tetanus-pertussis), DTaP-IPV-HepB
*Streptococcus* *pyogenes*	Gram-positive	Cases: 3729 ^1^ Deaths: 516 ^1^	phase I (NCT02564237)	Prevenar-13, Pneumovax-23, Synflorix
*Pseudomonas* *aeruginosa*	Gram-negative	Cases: n.a. Death: 300 ^1,4^	In vivo animal studies	-
*Listeria* *monocytogenes*	Gram-positive	Cases: 82 ^2,3^ Deaths: 132 ^2,3^	In vivo animal studies	-
*Chlamydia* *trachomatis*	Gram-negative	Cases: 128.5 ^1^ Deaths: n.a.	In vivo animal studies	-
*Yersinia pestis*	Gram-negative	Cases: n.a. Deaths: n.a.	In vivo animal studies	Vaccinum pestosum vivum
*Rhodococcus equi*	Gram-positive	Cases: n.a. Deaths: n.a.	In vivo animal studies	-

^1^ WHO, World Health Organization (2022–2023). ^2^ CDC, Centers for Disease Control and Prevention (2017–2023). ^3^ FDA, Food and Drug Administration (2020–2023). ^4^ Information was obtained on the basis of data from Refs. [15,18].

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
