# Peer review of "Current Progress in the Development of mRNA Vaccines Against Bacterial Infections"

_ijms, 2024, doi:10.3390/ijms252313139_

Round 1
Reviewer 1 Report
Comments and Suggestions for Authors
The manuscript ijms-3337504 entitled Current Progress in the Development of mRNA Vaccines against Bacterial Infections by Alina Khlebnikova , Anna Kirshina , Natalia Zakharova , Roman A. Ivanov , Vasiliy Reshetnikov discuss structural features and key mechanisms of evasion of an immune response for nine species of bacterial pathogens against which mRNA vaccines have been developed and tested in animals. They focused on results of experiments involving application of mRNA vaccines against various bacterial pathogens in animal models and discuss possible options for improving the vaccines’ effectiveness.
The discovery of the first antibiotics and the subsequent golden era of their discovery temporarily shifted the balance in this confrontation to the side of humans. Irrational use of antibacterial drugs and the evolution of bacteria got the better of humans again. Therefore, today, the search for new antibacterial drugs or the development of alternative approaches to the prevention and treatment of bacterial infections is relevant and topical again. Vaccination is one of the most effective strategies for the prevention of bacterial infections. The success of new-generation vaccines, such as mRNA vaccines, in the fight against viral infections has prompted many researchers to design mRNA vaccines against bacterial infections.
This review work is update and scientifically strong and gives to the reader a wide look to the field of antibacterial vaccinations.
The work is well written and interesting.
The two tables are informative.
English is very good
References are appropriated.
Other points:
Line 44: An alternative approach to preventing the spread of bacterial infections and to re- 44
ducing the associated mortality is vaccination.
Rewrite as:
Vaccination is an efficient and alternative approach to prevent the spread of bacterial infections and to reduce the associated mortality is vaccination.
Line 92: the tile of the Table 1 should stay together with the table
Line 240-251: the abbreviations can stay one after the other in few lines
Author Response
Dear Editor and Referee:
Thank you for allowing us to submit a revised draft of our manuscript titled “Current Progress in the Development of mRNA Vaccines against Bacterial Infections”. We are grateful for your review of the manuscript and your valuable comments and concerns. We have been able to incorporate into the manuscript most of the suggestions provided by the reviewers.
We have highlighted the revisions within the manuscript.
Below, marked in red, are point-by-point responses to the reviewers’ comments and concerns.
Sincerely,
Vasiliy Reshetnikov
Reviewer1:
The manuscript ijms-3337504 entitled Current Progress in the Development of mRNA Vaccines against Bacterial Infections by Alina Khlebnikova , Anna Kirshina , Natalia Zakharova , Roman A. Ivanov , Vasiliy Reshetnikov discuss structural features and key mechanisms of evasion of an immune response for nine species of bacterial pathogens against which mRNA vaccines have been developed and tested in animals. They focused on results of experiments involving application of mRNA vaccines against various bacterial pathogens in animal models and discuss possible options for improving the vaccines’ effectiveness.
The discovery of the first antibiotics and the subsequent golden era of their discovery temporarily shifted the balance in this confrontation to the side of humans. Irrational use of antibacterial drugs and the evolution of bacteria got the better of humans again. Therefore, today, the search for new antibacterial drugs or the development of alternative approaches to the prevention and treatment of bacterial infections is relevant and topical again. Vaccination is one of the most effective strategies for the prevention of bacterial infections. The success of new-generation vaccines, such as mRNA vaccines, in the fight against viral infections has prompted many researchers to design mRNA vaccines against bacterial infections.
This review work is update and scientifically strong and gives to the reader a wide look to the field of antibacterial vaccinations.
The work is well written and interesting.
The two tables are informative.
English is very good
References are appropriated.
Other points:
Line 44: An alternative approach to preventing the spread of bacterial infections and to re- 44
ducing the associated mortality is vaccination.
Rewrite as:
Vaccination is an efficient and alternative approach to prevent the spread of bacterial infections and to reduce the associated mortality is vaccination.
Line 92: the tile of the Table 1 should stay together with the table
Line 240-251: the abbreviations can stay one after the other in few lines
Reply: Thank you for your high appreciation of the manuscript, the text have been corrected.

Reviewer 2 Report
Comments and Suggestions for Authors
Good morning for all authors,
Analyzing with interest and attention the manuscript (Type Review) with ID: ijms-3337504-peer-review-v1, entitled "Current Progress in the Development of mRNA Vaccines against Bacterial Infections", Authors: Alina Khlebnikova†, Anna Kirshina*†, Natalia Zakharova, Roman A. Ivanov, Vasiliy Reshetnikov*, Section: Molecular Immunology, Special Issue: DNA and mRNA-Based Vaccines Against Infectious Diseases, for a possible publication in the prestigious Journal - IJMS,
I consider that:
1. The article's authors proposed a much-discussed topic in today's medical scientific world: the potency and/or the importance and efficiency of mRNA vaccines in bacterial infections.
2. The article follows all the specific instructions of the journal presented in aims and scope, instructions for authors, and other information about the journal IJMS.
3. Chapter 1. Introduction: the authors present relevant information for the chosen subject.
4. Chapter 2. Statistics on cases of illnesses and deaths due to infections caused by bacteria: data presented in all nine subchapters are very well structured and coherent. - In Tables 1-2, all the data are presented with clarity; thus, they can be easily analyzed and interpreted!
5. Chapter 3. In silico approach: the authors exemplified and compared all the results obtained in different studies and/or assessments of other authors, according to the specified bibliography.
6. Chapter 4. Discussion and conclusion: are very well structured and coherent.
7. The bibliography chosen by the authors corresponds to the requirements and refers to the subject of this article.
8. All authors have made a fair contribution to the study.
9. The authors also received funding for this study (agreement No. 075-10-2021-113, unique project ID RF-193021×0001) demonstrating professionalism and medical scientific relevance.
In conclusion:
I ACCEPT for publication in the International Journal of Molecular Sciences (IJMS, MDPI, ISSN: 1422-0067, IF=4.9) this article (Type Review) with ID: ijms-3337504-peer-review-v1, entitled "Current Progress in the Development of mRNA Vaccines against Bacterial Infections", Authors: Alina Khlebnikova†, Anna Kirshina*†, Natalia Zakharova, Roman A. Ivanov, Vasiliy Reshetnikov*, Section: Molecular Immunology, Special Issue: DNA and mRNA-Based Vaccines Against Infectious Diseases.
Congratulations to all the authors of this article!

Author Response
Dear Editor and Referee:
Thank you for allowing us to submit a revised draft of our manuscript titled “Current Progress in the Development of mRNA Vaccines against Bacterial Infections”. We are grateful for your review of the manuscript and your valuable comments and concerns. We have been able to incorporate into the manuscript most of the suggestions provided by the reviewers.
We have highlighted the revisions within the manuscript.
Below, marked in red, are point-by-point responses to the reviewers’ comments and concerns.
Sincerely,
Vasiliy Reshetnikov
Reviewer 2
Good morning for all authors,
Analyzing with interest and attention the manuscript (Type Review) with ID: ijms-3337504-peer-review-v1, entitled "Current Progress in the Development of mRNA Vaccines against Bacterial Infections", Authors: Alina Khlebnikova†, Anna Kirshina*†, Natalia Zakharova, Roman A. Ivanov, Vasiliy Reshetnikov*, Section: Molecular Immunology, Special Issue: DNA and mRNA-Based Vaccines Against Infectious Diseases, for a possible publication in the prestigious Journal - IJMS,
I consider that:
- The article's authors proposed a much-discussed topic in today's medical scientific world: the potency and/or the importance and efficiency of mRNA vaccines in bacterial infections.
- The article follows all the specific instructions of the journal presented in aims and scope, instructions for authors, and other information about the journal IJMS.
- Chapter 1. Introduction: the authors present relevant information for the chosen subject.
- Chapter 2. Statistics on cases of illnesses and deaths due to infections caused by bacteria: data presented in all nine subchapters are very well structured and coherent. - In Tables 1-2, all the data are presented with clarity; thus, they can be easily analyzed and interpreted!
- Chapter 3. In silico approach: the authors exemplified and compared all the results obtained in different studies and/or assessments of other authors, according to the specified bibliography.
- Chapter 4. Discussion and conclusion: are very well structured and coherent.
- The bibliography chosen by the authors corresponds to the requirements and refers to the subject of this article.
- All authors have made a fair contribution to the study.
- The authors also received funding for this study (agreement No. 075-10-2021-113, unique project ID RF-193021×0001) demonstrating professionalism and medical scientific relevance.
In conclusion:
I ACCEPT for publication in the International Journal of Molecular Sciences (IJMS, MDPI, ISSN: 1422-0067, IF=4.9) this article (Type Review) with ID: ijms-3337504-peer-review-v1, entitled "Current Progress in the Development of mRNA Vaccines against Bacterial Infections", Authors: Alina Khlebnikova†, Anna Kirshina*†, Natalia Zakharova, Roman A. Ivanov, Vasiliy Reshetnikov*, Section: Molecular Immunology, Special Issue: DNA and mRNA-Based Vaccines Against Infectious Diseases.
Congratulations to all the authors of this article!
Reply: Thank you for your high appreciation of the manuscript!

Reviewer 3 Report
Comments and Suggestions for Authors
Pointed Suggestions for Improvement
Abstract
- Specify the unique contributions of the review in comparison to existing literature.
- Use precise terms, avoiding broad statements like "irrational use of antibacterial drugs."
Introduction
- Highlight the novelty of focusing on mRNA vaccines against bacterial infections.
- Add brief context on how bacterial infections differ in their immunological challenges compared to viral infections.
Table Presentation
- in Table 2, highlight which bacteria show the most promise for mRNA vaccine development
Section Organization
- Merge overlapping sections, such as those on adaptive immunity and vaccine effectiveness, for smoother reading.
- Ensure subheadings align with the content's focus (e.g., split "Vaccines against M. tuberculosis" into distinct subsections for types of vaccines, trial stages, and results).
Please rephrase
- "mRNA vaccines allow an investigator to 'tune' an optimal ratio of humoral to cellular immunity"
- Maintain consistency in technical terms (e.g., "Gram-positive" versus "Gram-positive bacteria").
Discussion of Results
- Expand on the broader implications of in vivo results for human clinical applications.
- Discuss limitations or challenges (e.g., delivery systems, immune evasion mechanisms) in more depth.
Figures and Visual Aids
- For Figure 2, explain the relevance of observed immune responses to vaccine development.
Future Directions
- Include suggestions for optimizing delivery systems for bacterial mRNA vaccines.
- Discuss the potential of combining mRNA vaccines with existing antibiotic therapies.
Conclusion
- Strengthen the conclusion by summarizing actionable insights:
- Which bacterial pathogens are the most promising candidates for mRNA vaccine development?
- Key areas for further research.
References
- Include more recent studies (2023–2024) to bolster the review’s relevance.
Author Response
Dear Editor and Referee:
Thank you for allowing us to submit a revised draft of our manuscript titled “Current Progress in the Development of mRNA Vaccines against Bacterial Infections”. We are grateful for your review of the manuscript and your valuable comments and concerns. We have been able to incorporate into the manuscript most of the suggestions provided by the reviewers.
We have highlighted the revisions within the manuscript.
Below, marked in red, are point-by-point responses to the reviewers’ comments and concerns.
Sincerely,
Vasiliy Reshetnikov
Reviewer 3.
Abstract
- Specify the unique contributions of the review in comparison to existing literature.
Use precise terms, avoiding broad statements like "irrational use of antibacterial drugs."
Reply: Thank you, the text has been corrected.
Introduction
- Highlight the novelty of focusing on mRNA vaccines against bacterial infections. Add brief context on how bacterial infections differ in their immunological challenges compared to viral infections.
Reply: Thank you for highlighting that. We have added to the introduction a novelty and description of the differences between bacterial and viral infections that may be encountered in the development of an mRNA vaccine.
Table Presentation
- in Table 2, highlight which bacteria show the most promise for mRNA vaccine development
Reply: We discussed the potential for human clinical use of mRNA vaccines for which in vivo data have been obtained in 5.Conclusion.
Section Organization
- Merge overlapping sections, such as those on adaptive immunity and vaccine effectiveness, for smoother reading.
Reply: Thank you for your comment, but we believe that separating results of adaptive immunity from protective immunity is appropriate. Adaptive and protective responses are interrelated, but the ability of a vaccine to induce an adaptive response does not always imply the presence of an adequate protective response. Evaluation of a vaccine for a protective response is the next step after the presence of an adaptive response. The ultimate goal of vaccination is to achieve an effective protective response.
- Ensure subheadings align with the content's focus (e.g., split "Vaccines against M. tuberculosis" into distinct subsections for types of vaccines, trial stages, and results).
Reply: We have added subsections such as “Features of the structure and immune evasion” to the description of each bacterium, making the sections more structured and easier to read. We have also divided the section about vaccines into “non-mRNA vaccines” and “mRNA vaccines”. We believe that a more specific division is unnecessary.
Please rephrase
- "mRNA vaccines allow an investigator to 'tune' an optimal ratio of humoral to cellular immunity"
- Maintain consistency in technical terms (e.g., "Gram-positive" versus "Gram-positive bacteria").
Reply 6-7. Thank you for pointing this out. We rephrased and corrected the text.
Discussion of Results
- Expand on the broader implications of in vivo results for human clinical applications. Discuss limitations or challenges (e.g., delivery systems, immune evasion mechanisms) in more depth.
Reply: We have taken this comment into account. We have extended analysis of the limitations and challenges associated with the development of bacterial mRNA vaccines. We discussed the potential for human clinical use of mRNA vaccines for which in vivo data have been obtained in 5.Conclusion.
Figures and Visual Aids
- For Figure 2, explain the relevance of observed immune responses to vaccine development.
Reply: This review does not include a second figure due to the large volume of the manuscript. Therefore, we believe that adding a second figure would be unnecessary
Future Directions
- Include suggestions for optimizing delivery systems for bacterial mRNA vaccines. Discuss the potential of combining mRNA vaccines with existing antibiotic therapies.
Reply: We discussed the potential for human clinical use of mRNA vaccines for which in vivo data have been obtained.
Conclusion
- Strengthen the conclusion by summarizing actionable insights: Which bacterial pathogens are the most promising candidates for mRNA vaccine development? Key areas for further research.
Reply: We have added a Conclusion section based on your suggestions.
References
- Include more recent studies (2023–2024) to bolster the review’s relevance.
Reply: We have added a few recent studies in reference list.
